# Primary cilia in osteoblasts and osteocytes are required for skeletal development and mechano-adaptation

Mariana Moraes de Lima Perini[1], Alyssa F. Fayemi[1], Julie N. Pugh[1], Elizabeth M. Scott[2], Karan Bhula[2], Austin Chirgwin[2], Olivia N. White[2], Nicolas F. Berbari[1], Jiliang Li[1]*

1 Department of Biology, Indiana University, Indianapolis, Indiana, 2 Department of Biomedical Engineering, Purdue University, Indianapolis, Indiana

* jilili@iu.edu

## Abstract

Primary cilia play a crucial role in the development and mechanosensation of various tissue types, including bone. In this study, we investigated their role in bone growth and adaptation by targeting two cilia specific genes, IFT88 and MKS5. Conditional knockout (cKO) of IFT88 in osteoblasts significantly reduced body weight and femur length in mice compared to the littermate controls. Additionally, female IFT88 cKO mice exhibited a significant suppression of bone formation rate compared to the littermate controls. To further explore the role of primary cilia in osteocytes, osteocytes specific MKS5 cKO mice underwent axial ulnar loading at a peak force of 2.9N for females and 3.2N for males with 120 cycles per day for three consecutive days. Load induced bone formation rate was significantly decreased by 48% in males and 42% in females compared to the littermate control mice. These findings underscore the critical role of primary cilia in bone development and mechano-adaptation. They suggest that functional primary cilia in osteoblasts are essential for skeletal development, while those in osteocytes mediates mechanically induced bone formation, highlighting its potential as therapeutic targets for bone loss prevention.

## Introduction

The primary cilium is a single, immotile, microtubule-based organelle found on nearly every cell in the body. It has emerged as a critical regulator of multiple signaling pathways [1–4]. The primary cilium is indispensable for development, and plays a key role in mechanosensing across various tissue types, including the liver, kidney, endothelium, and bone [2]. Defects or dysfunctions in primary cilia have been linked to severe multisystemic disorders, underscoring their biological significance [3,5].

The assembly of the primary cilium depends on the transport of axonemal precursors, including α-tubulin and β-tubulin, from the cell body to the ciliary tip [5]. While many proteins localize to the primary cilium, they are not synthesized there and

**Data availability statement:** Figshare: 10.6084/m9.figshare.31044895.

**Funding:** This project was supported by National Institute of Health R21 AR074012.

**Competing interests:** The authors have declared that no competing interests exist.

require a specialized transport system known as intraflagellar transport (IFT) [4]. IFT is essential for both assembling and maintenance of the primary cilium and is divided into two complexes, A and B, which work alongside motor proteins to shuttle ciliary components. Complex A, composed of six IFT proteins and dynein motor proteins (DYNC2H1 and DYNC2L1), regulates the retrograde transport (tip to base) [4,5]. Complex B, consisting of approximately 14 IFT proteins, including IFT88 governs anterograde transport (base to tip) via Kinesin-II motor proteins (KIF3A, KIF3B, and KAP3). IFT88, previously known as Tg737 or Polaris, encodes a key component of the IFT-B complex. Mutations disrupting IFT88 or other IFT components impair primary cilium formation, likely due to the failure of ciliary proteins to be transported via the anterograde complex [6,7].

The transition zone (TZ) of cilia, located between the basal body of the cell and the cilium, plays a crucial role in regulating the selective entry and exit of proteins [8,9]. Several protein complexes localize within TZ, including the Meckel Syndrome (MKS) and Nephronophthisis (NPHP) complexes. Retinitis pigmentosa GTPase regulator-interacting protein 1-like (RPGRIP1L), also known to as MKS5, Ftm, and Nphp8, function as an anchor between the cilium and the basal body, controlling ciliary gating [10,11]. MKS5 is thought to be a key regulator of the TZ, acting as a "bridge" between the MKS and NPHP modules. Additionally, it may function as an assembly factor for all other MKS proteins and contribute to the formation of the TZ structure [8, 11]. Mutations affecting proteins within both in IFT complexes and the TZ structure can result in absent or dysfunctional cilia by disrupting ciliary membrane composition, mislocalizing signaling proteins and destablizing other TZ components [8,9,11,12].

Growing evidence indicates that the primary cilium is essential for bone homeostasis and mechanotransduction. Deletion of IFT88 in chondrocytes reduced ciliation, disrupted chondrocyte differentiation and impaired growth plate ossification, leading to skeletal abnormalities [13]. Additionally, primary cilia dysfunction induced by TGF-β1 suppressed human osteoblast responses to fluid flow, a key mechanical stimulus for bone adaptation [14]. Interestingly, this impairment can be reversed with Tubacin treatment, an HDAC6 inhibitor, suggesting potential therapeutic strategies for restoring cilia function [14].

Moreover, osteocytes with elongated primary cilia induced by fenoldopam and lithium exhibited enhanced mechanosensitivity, highlighting the importance of ciliary length in mechanotransduction [15]. Conditional knockout of the *IFT88* in osteochondroprogenitors, driven by *Prx1-Cre*, severely reduced mechanically induced bone formation and impaired the mineralization necessary for daily skeletal maintenance in mice, reinforcing the critical role of cilia in bone adaptation [16]. However, it remains unclear whether disrupting primary cilia formation in more mature, osteoblast-committed cells using *Osterix-Cre* would produce similar effects on bone development and remodeling. Understanding these mechanisms could provide new insights into bone diseases and potential therapeutic targets.

Osteocytes play a central role in regulating the balance between bone formation and resorption. Disruptions in mechanotransduction contribute to metabolic bone disorders, such as osteoporosis. Understanding how primary cilia mediate osteocyte

responses to mechanical loading could help identify novel molecular targets for osteoporosis prevention and treatment. A previous doctoral thesis has reported that targeted deletion of IFT88 by breeding DMP-1 cre mice and IFT88 floxed mice did not affect skeletal development but did inhibit cilia formation in osteocytes [17]. The load-induced bone formation was significantly reduced by 47% in the conditional knockout mice compared to the controls [17]. Instead of eliminating primary cilia in osteocytes, we investigated mice with dysfunctional cilia due to MKS5 disruption. MKS5 primarily functions at the ciliary transition zone, and its mutations are linked to ciliopathies, including Meckel-Gruber syndrome (MKS). MKS is characterized by skeletal and neurodevelopmental abnormalities such as polydactyly, cognitive impairment and retinal degeneration.

In this study, we investigated the role of primary cilia in bone growth and mechao-adaptation by targeting IFT88 in osteoblasts and MKS5 in osteocytes. The use of two distinct ciliary genes - *IFT88* in osteoblast-lineage cells and *MKS5* in osteocytes – was intentional and reflects the stage-specific roles of the primary cilium in bone biology. *IFT88* is a core component of the intraflagellar transport (IFT) machinery and is essential for primary cilium assembly [1,9,17–19]. Targeting *IFT88* in osteoblast-committed cells allows direct evaluation of how complete loss of cilium formation affects postnatal skeletal development. In contrast, *MKS5* functions at the transition zone at the base of the primary cilium, regulating ciliary gating and signaling rather than ciliogenesis itself [8,11]. Disruption of *MKS5* results in structurally present but functionally impaired cilia [1,10], providing a model to study altered ciliary signaling in long-lived, terminally differentiated osteocytes without eliminating the organelle entirely. This approach minimizes developmental confounding effects while preserving osteocyte survival and network integrity. Together, the complementary use of *IFT88* and *MKS5* enables cell-type–specific interrogation of primary cilium function across different stages of osteoblast lineage differentiation, allowing us to distinguish the roles of cilium assembly versus ciliary signaling in bone growth and mechano-adaptation. These findings highlight the critical role of primary cilia in skeletal integrity and their potential as therapeutic targets for bone disorders.

## Materials and methods

### Animal models

All animal procedures were reviewed and approved by the Institutional Animal Care and Use Committee (IACUC) at the Indiana University Indianapolis.

Osteoblast-specific deletion of the IFT88 gene was achieved with the breeding of Osx1-GFP::Cre (Stock No. 006361, Jackson Laboratories, Bar Harbor, ME, USA) positive mice and mice with LoxP sites flanking the IFT88 exon (Ift88$^{LoxP/LoxP}$, Stock No. 022409, Jackson Laboratories, Bar Harbor, ME, USA) [20], generating Osx-Cre; IFT88$^{LoxP/LoxP}$ (IFT88cKO mice), and Osx-Cre; IFT88$^{+/+}$ (littermate control). A total of 31 Cre positive (+) mice were collected at eight weeks of age, seven male IFT88$^{+/+}$ (M Control), eight male IFT88$^{LoxP/LoxP}$ (M IFT88cKO), eight female IFT88$^{+/+}$ (F Control), and eight female IFT-88$^{LoxP/LoxP}$ (F IFT88cKO).

Dmp1–8kb-Cre mice possessing the transgene orienting the Cre recombinase sequence driven by the murine dentin matrix protein 1 (Dmp1) gene [21] were bred with MKS5$^{LoxP}$ mice, generating Dmp1-Cre; MKS5 $^{+/LoxP}$ mice that were crossed with the purpose of producing Dmp1-Cre; MKS5$^{LoxP/LoxP}$ (MKS5cKO) mice and the littermate controls, Dmp1-Cre; MKS5$^{+/+}$ [20,22], in order to obtain mice with osteocyte-specific deletion of the MKS5 gene. A total of 29 Cre (+) mice were collected at 20 weeks of age, eight male MKS5$^{+/+}$ (M Control), eight male MKS5$^{LoxP/LoxP}$ (M MKS5cKO), seven female MKS5$^{+/+}$ (F Control), and six female MKS5$^{LoxP/LoxP}$ (F MKS5cKO).

The genotype of each mouse was determined and confirmed by polymerase chain reaction (PCR) amplification of tail DNA samples at weaning and sacrifice. All primers were acquired through IDT (Integrated DNA Technologies, Inc., Skokie, IL, USA). When genotyping for Osx-Cre, the primer sequences used for its amplification were as follows: Osx-Cre forward (TGCK) – 5' – CTC TTC ATG AGG AGG ACC CT −3', Osx-cre reverse (OSX) – 5' – GCC AGG CAG GTG CCT GGA CAT −3'. For the amplification of IFT88 LoxP, the primers used were IFT88 Common forward – 5' – GCC TCC TGT TTC TTG ACA ACA GTG – 3', IFT88 LoxP and WT Reverse – 5' – GGT CCT AAC AAG TAA GCC CAG TGT T – 3', and IFT88 Null

Reverse – 5'- CTG CAC CAG CCA TTT CCT CTA AGT CAT GTA – 3'. When genotyping for Osx-Cre, a 615-bp band indicated the presence of the recombinase, while the lack of a band indicated its absence. When genotyping for IFT88 LoxP, a 350-bp band indicated IFT88$^{flox/flox}$ individuals, while a 310-bp band signified IFT88$^{+/+}$ individuals. Heterozygous individuals were signified by the presence of three bands, at 350-bp, 310-bp, and 280-bp.

The primer sequences used for the amplification of Dmp1-Cre were Dmp1-Cre forward (CsCre1) – 5' – AGG GAT CGC GAG GCG TTT TC – 3' and Dmp1-Cre reverse (CsCre2) – 5' – GTT TTC TTT TCG GAT CCG CC –3'. For the amplification of MKS5 LoxP, there were three primer sequences used. They were MKS5 Common forward – 5' – GTC CTC TGA CTT CCA GTG TCA TGT GC – 3', MKS5 LoxP and WT Reverse – 5' – GTG GGT TGT ACA GTT TCT GCT TCA TCC AC – 3', and MKS5 Null Reverse – 5' – AAG CTC TAA AGC TGG GAC TGC AGC – 3'. When genotyping for Dmp1Cre, the presence of a 300-bp band indicated the presence of Dmp1-Cre recombinase, while the lack of a band indicated its absence. When genotyping for MKS5 LoxP, one single 788-bp band indicated MKS5$^{flox/flox}$, while one single 605-bp band indicated MKS5$^{+/+}$. The presence of both bands together indicated heterozygous individuals with only one LoxP site. The heterozygous mice from both the MKS5 and the IFT88 lineages were only used for breeding purposes.

### In Vivo Ulnar Loading

In vivo axial ulnar loading was performed on 20-week-old (± 14 days) Dmp1-Cre; MKS5$^{LoxP}$ mice and their littermate controls. The animals were subjected to axial ulnar loading for 3 consecutive days as previously published [23]. The right ulnae were loaded for 120 cycle/day at a frequency of 2 Hz with a peak force of 2.9 N for female mice and 3.2 N for male mice. The left ulnae served as internal controls. All animals were allowed regular cage activities following loading sessions. Two days following the final day of loading (10 days prior to sacrifice), the animals were injected with 0.4 mL green fluorochrome label calcein (30 mg/Kg body weight) via intraperitoneal injection, and 7 days later (3 days prior to sacrifice) they were administered 0.4 mL of red fluorochrome label alizarin (50 mg/Kg body weight) via intraperitoneal injection. All test animals were sacrificed 3 days after the administration of alizarin by $CO_2$ gas followed by cervical dislocation in order to assure death.

### Histology and Histomorphometry

The Osx-Cre; IFT88 mice were injected with green fluorochrome label calcein (30 mg/Kg body weight) via intraperitoneal injection at seven weeks of age. Then, 5 days later (2 days prior to sacrifice) they were administered red fluorochrome label alizarin (50 mg/Kg body weight) via intraperitoneal injection. All test animals were sacrificed 2 days after the administration of alizarin, at eight weeks of ageby $CO_2$ gas followed by cervical dislocation in order to assure death.

For the Osx-Cre; IFT88 animals, the right femur was processed for histological and histomorphometric analysis. For the Dmp1-Cre; MKS5 test groups, the right distal femur and right and left ulnae were processed for histological and histomorphometric analysis. All samples were processed following the protocol previously established [23].

The bones were placed in a vial containing 10 mL of 10% neutral-buffered formalin (NBF) and stored in 4°C for 24 hours. The formalin was then replaced by 70% ethanol (EtOH) and placed in the 4°C storage until ready to be processed. The bones were dehydrated by a series of graded alcohols with the concentration increased every 2–8 hours until it reached 100% EtOH. The 100% EtOH was replaced by a 1:1 mixture of EtOH and methyl methacrylate (MMA) for 2–8 hours. Then the samples were placed in undiluted MMA for 2–8 hours, which was later replaced by infiltration media (MMA and 4% dibutyl phthalate) and placed under vacuum for 2–7 days at 17 inHg.

Using a wire saw, the ulna was cut proximal from the pencil mark drawn at the midshaft during tissue processing to ensure midshaft collection. Then 5 sequential transverse thick sections (50μm) were collected. The sections were mounted on microscope slides using Eukitt® mounting medium (MilliporeSigma, St. Louis, MO, USA) and xylenes-thinned mounting medium were applied to the mounted section, increasing the surface area for grinding. The slides were ground to 30–50 μM against wet 600 grit sandpaper, then they were cover slipped using xylenes-thinned mounting medium.

MMA blocks embedding the distal femur were cut into thin sections on a rotary microtome (Leica Biosystems Inc., Buffalo Grove, IL) using a tungsten carbide knife. The block was secured on the microtome and 50% EtOH was applied to the face of the block for softening. Five microscope slides previously subbed with 1% gelatin solution (Thermo Fisher Scientific, Waltham, MA, USA) were prepared, each containing three 4 µm thick sections. The slides were left to dry at 37ºC overnight, then cover slipped.

Quantitative analysis of bone samples was performed using OsteoMeasure™ histomorphometry software (Osteo-Metrics, Decatur, GA, USA). The slides were analyzed using an X-Cite Series 120Q fluorescence illuminator (Excelitas Technologies, Waltham, MA, USA) on an Olympus DP72 digital microscope camera connected to an Olympus BX53 Tele-pathology Microscope System (Olympus Scientific Solutions Americas, Waltham, MA, USA). Measurements were taken by tracing the sections at 200x magnification on a touchscreen (Wacom, Kazo, Saitama Prefecture, Japan).

Unstained cortical sections of the left and right ulnae, as well as an unstained right femur midshaft cortical section of each study subject were analyzed for periosteal perimeter, endocortical perimeter, periosteal single label, endocortical single label, periosteal double label, and endocortical double label. Using these tracings, derived measurements, such as MS/BS, MAR, and BFR/BS, were calculated. As the left ulna was used as a non-loaded control, the values for the left ulna were subtracted from the values for the right ulna, resulting in the amount of bone activity occurring because of loading.

One slide of the distal femur of each test subject was stained with tartrate-resistant acid phosphatase (TRAP) and counterstained with a toluidine blue. Histomorphometry was performed using OsteoMeasure™ histomorphometry software (OsteoMetrics, Decatur, GA, USA). The slides were analyzed under bright-field illumination on an Olympus DP72 digital microscope camera connected to an Olympus BX53 Telepathology Microscope System (Olympus Scientific Solutions Americas, Waltham, MA, USA). Each sample was analyzed for osteoclast number (N.Oc), osteoclast surface per bone surface (Oc.S/BS), osteoclast number over bone perimeter (N.Oc/B.Pm), and osteoclast number over perimeter (N.Oc/Oc.Pm).

## Peripheral Dual-Energy X-Ray Absorptionmetry (PIXImus)

The bone mineral density (BMD, g/cm$^2$) and mineral content (BMC, g) of the left femur of each test animal were analyzed using a PIXImus dual-energy X-ray absorptiometer (DXA) (PIXImus Lunar Corp., Madison, WI, USA).

## Micro-Computed Tomography (µCT)

The left femur of each test animal was imaged with a SkyScan 1172 micro-computed tomography (µCT) (SkyScan, Kontich, Belgium) following the protocol previously established [24]. Reconstruction of the 3D structures was done through NRecon reconstruction software (Micro Photonics Inc., Allentown, PA, USA). CTan software was used to select regions of interest (ROI) for each midshaft. Cortical ROIs composed of 5 images each, averaging 10 µm of cortical bone surface was utilized in this analysis. The trabecular ROI was defined as 1 mm of the trabecular region of the distal femur beginning 1 mm proximal to the growth plate.

## Biomechanical Testing

The left femurs were tested in three-point bending tests to failure to evaluate structural-level and tissue-level mechanical properties. The support span was set at 5.4 mm and 8 mm for Osx-Cre;Ift88 and Dmp1-Cre;MKS5, respectively. Each femur was oriented so the loading point was applied at the 50% region of the bone and the anterior surface of the bone was in tension. Each test was conducted under displacement control at a loading rate of 0.025 mm/sec until failure. During each test, the resulting force and displacement were recorded.

Structural-level mechanical properties were obtained from collected force-displacement data and included ultimate force, yield force, total work and stiffness. To calculate tissue-level mechanical properties of the bones, the recorded force and displacement data were converted to stress and strain using geometric properties obtained from µCT analysis.

A region of interest of 10 transverse slices was obtained from µCT scans at the 50% region of each bone and used to calculate the necessary geometric properties. These geometric properties were then used along with the force and displacement data in a custom MATLAB (MathWorks, Natick, MA) script, from Professor Joseph Wallace at Purdue University Indianapolis, Department of Biomedical Engineering (BME), to calculate stress and strain using standard engineering equations to then obtain tissue-level mechanical properties. The collected structural-level mechanical properties included ultimate stress, Young's modulus and toughness [25].

### Imaging of Primary Cilia

Serial digestion of calvariae from newborn Osx-Cre;IFT88 mice (1–5 days old) was performed to isolate primary bone cells. Calvariae were dissected and cleaned from any soft tissue and placed in media (α-MEM (minimum essential media), 10% FBS (fetal bovine serum), 1% P/S (penicillin-streptomycin antibiotic)) while PCR and gel electrophoresis were run to confirm the genotype. Once the genotype had been determined, the calvariae were cut into 3–4 pieces and placed in conical tubes with 3 mL of digest solution, composed of 0.1% collagenase, 0.05% trypsin, 4 mM ethylenediaminetetraacetic acid (EDTA). The tubes were then placed on a shaker for 20 minutes. The supernatant was removed, filtered, and transferred to another conical tube. This process was repeated 4 times, and after the last one, the new conical tubes containing the supernatant were centrifuged. The cells were plated and left to grow.

Once the cells were confluent, they were starved in media containing α-MEM, 0.1% FBS and 1% P/S for 24 hours to promote primary cilium growth. Cells were fixed with 4% paraformaldehyde (PFA) and 10% sucrose solution for 10 minutes at room temperature. The solution was then aspirated, the cells washed with 1x phosphate buffered saline (PBS), and 100% Methanol was added to the cells for 20 minutes. Once the Methanol was aspirated off, 1x PBS with 0.1% Triton X was added and the plate was incubated for 6 minutes. The cells were then washed with 1x PBS again, and a blocking solution (PBS+, composed of 2% donkey serum, 0.3% Triton X-100, 1% Bovine Serum Albumin, 10X PBS, and 0.02% Sodium Azide) was added after. The cells were left in room temperature with the blocking solution for one hour. The primary antibody (Monoclonal Anti-Acetylated Tubulin, Sigma T7451 and Monoclonal Anti-g-Tubulin, Sigma T6557) was then added, and the cells were incubated overnight. The cells were once again washed with PBS+, and the secondary antibody (Alexa Fluor™ 488 donkey anti-mouse IgG (H+L), Invitrogen #A21202 and Alexa Fluor™ 594 goat anti-rabbit IgG (H+L)) was added following the wash for one hour. The cells were then washed in PBS+ and Hoechst (Hoechst 33342, ThermoFisher Scientific #62249) was added in a 1:1000 dilution. Imaging was performed using a Keyence semi-automated microscope.

### Statistical Analysis

The data are expressed as mean ± SD (standard deviation). Group differences were analyzed using Welch's One-Way Analysis of Variance (ANOVA), followed by post hoc unpaired Welch's t-tests for pairwise comparisons, using GraphPad Prism (GraphPad Software, San Diego, CA). Significance was defined as $p < 0.05$.

## Results

### Abrogation of primary cilia in osteoblasts lead to reduced body weight and bone size in IFT88cKO mice

In order to confirm the cilia phenotype in the IFT88cKO mice, osteoblast cells were isolated from the calvaria of newborn pups. Immunocytochemistry was utilized to image primary cilia (anti-acetylated alpha-tubulin). Primary cilia were found on cells from littermate control mice, while they were absent from cells of IFT88cKO mice (Fig 1).

A significant decrease in body weight was observed in both male (male control 20.51g ± 1.168, IFT88cKO male 18.42g ± 1.1909, p = 0.0048) and female (female control 17.736g ± 2.796, IFT88cKO female 15.655g ± 1.824, p = 0.0374) groups when comparing IFT88cKO mice to the littermate controls (Fig 2A & B). The femurs of the IFT88cKO mice were significantly shorter than the femurs of the littermate control mice of both male (male control 1.457 mm ± 0.053, IFT88cKO

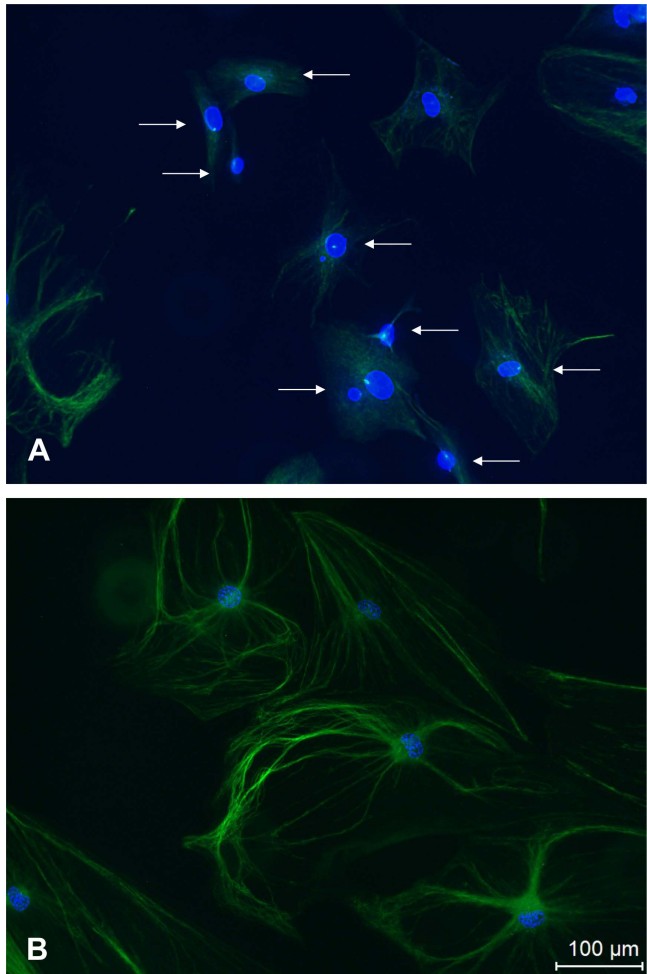

**Fig 1. Representative images of primary cilia in osteoblasts isolated from (A) in Osx-Cre; IFT88⁺/⁺ and (B) Osx-Cre; IFT88ᴸᵒˣᴾ/ᴸᵒˣᴾ mice.** Cells isolated from newborn Osx-Cre; IFT88⁺/⁺ and Osx-Cre; IFT88ᴸᵒˣᴾ/ᴸᵒˣᴾ mice and cultured in vitro. Bright Green (white arrows): acetylated α-tubulin, a common ciliary marker was observed in **(A)** Osx-Cre;IFT88+/+osteoblasts, but not in **(B)** Osx-Cre; IFT88ᴸᵒˣᴾ/ᴸᵒˣᴾ osteoblastsGreen: gamma-tubulin. Blue: Hoechst, used to stain nuclear DNA.

male 1.31 mm ± 0.074, p = 0.0003) and female (female control 1.428 mm ± 0.0488, IFT88cKO female 1.3 mm ± 0.0756, p = 0.0019) groups (**Figure 2C1, C2 & D**). No significance was observed in the BMD or BMC of either group (Fig 2E & F). These data suggest the loss of primary cilia in osteoblasts suppresses bone growth in mice.

Trabecular analysis of µCT scans found that female IFT88cKO mice had a significant increase in Tb. N (p = 0.0443) when compared to female control mice (**Fig 3**A3). Furthermore, histological analysis of the thin trabecular sections with calcein-alizarin red double labeling revealed a significant decrease in MAR (p = 0.0005) and BFR/BS (p < 0.0001) of female IFT88cKO mice when compared to the female control (**Fig 3**B2 & B4).

Although it did not reach significance, the male conditional knockout group had a MAR that was 57.57% lower and a BFR/BS rate that was 58.03% lower compared to the littermate control group (**Fig 3**B2 & B4).

TRAP positive osteoclasts were counted on thin sections of the distal femur to estimate bone resorption. When compared between the littermate control and cKO groups, no significance in Oc. N/BS or Oc.S/BS was observed (Fig 3**C1 - C6**).

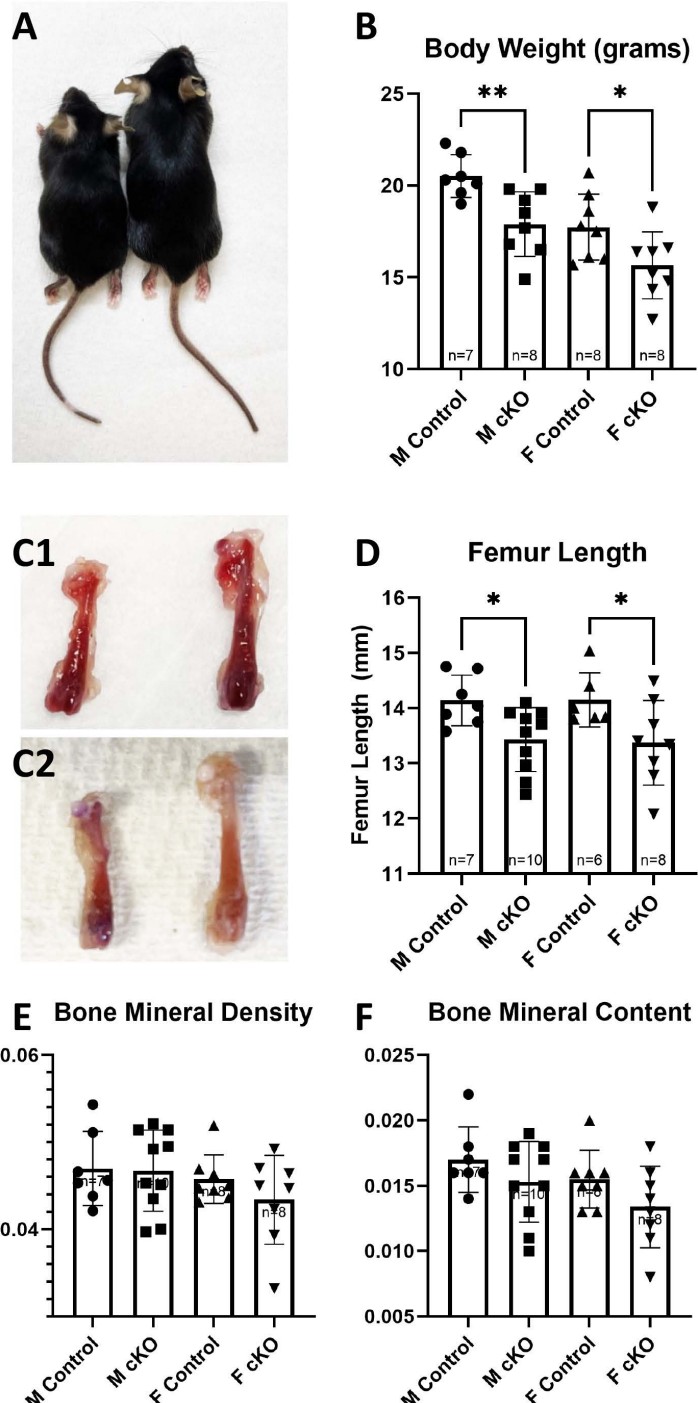

**Fig 2. Body weight, femoral length, BMD and BMC of Osx-Cre; IFT88$^{LoxP/LoxP}$ (cKO) mice and the littermate controls. (A)** Photo example of difference in size between the contral and IFT88cKO mice. **(B)** Body weights of the control and IFT88cKO mice at 8 weeks of age. Animals of IFT88cKO groups had significantly lower body weights compared to the control animals. (C1) Photo example of difference in length of femur between male IFT88cKO (Left) and the control (Right). (C2) Photo example of difference in length of femur between female IFT88cKO (Left) and the control (Right). **(D)** Length of femur of the control and IFT88cKO mice at 8 weeks of age. Male and female IFT88cKO had significantly shorter femurs when compared to the littermate controls. Values for **(E)** BMD and **(F)** BMC. PIXImus scans of femurs reveal no difference in either BMD or BMC of IFT88cKO animals compared to the controls. Note that *: p<0.05, **: p<0.01.

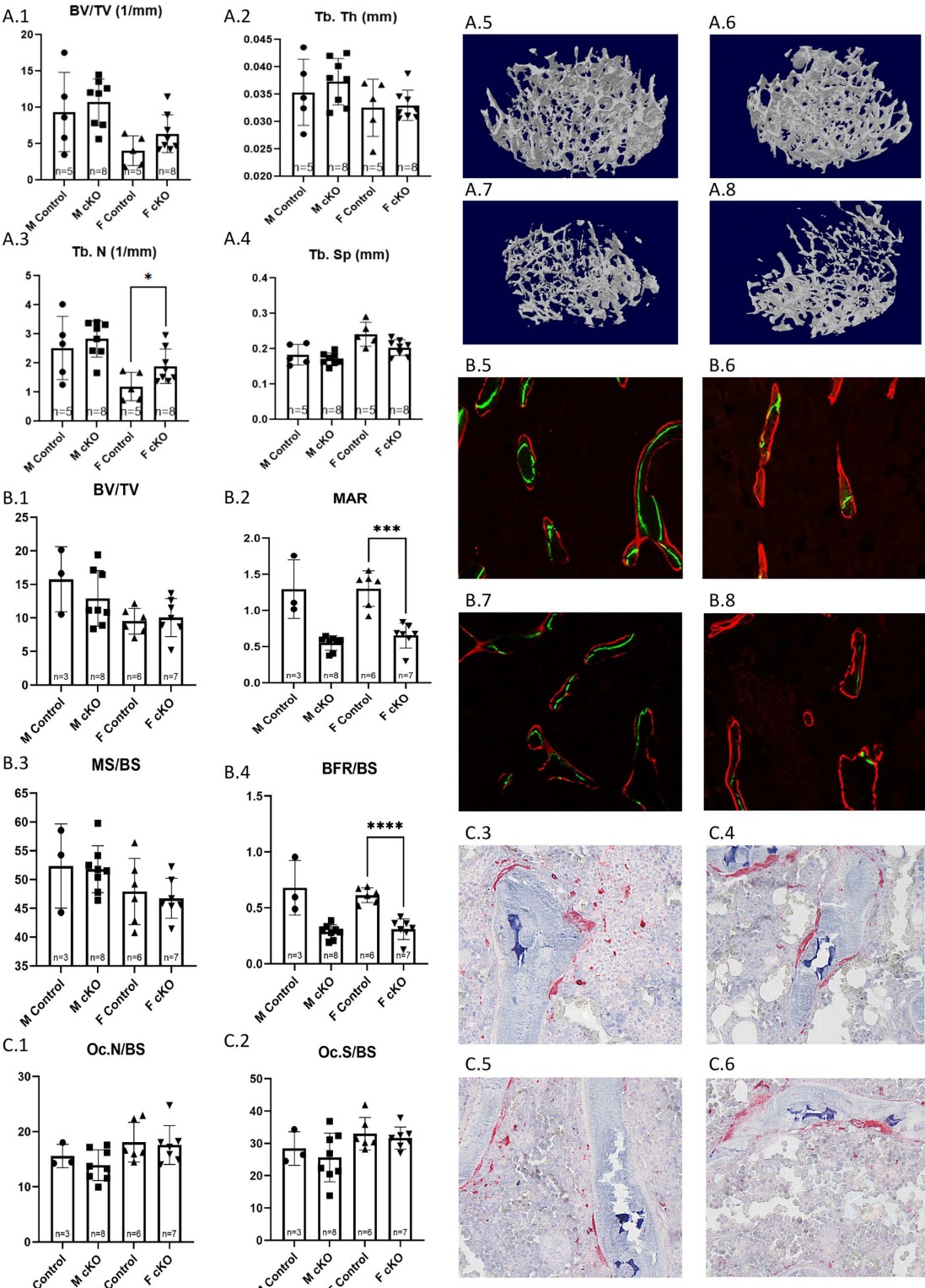

**Fig 3. Trabecular analysis of Osx-Cre, IFT88<sup>LoxP/LoxP</sup> (cKO) Mice.** (A1-4) Trabecular analysis results for distal femur of male and female IFT88cKO mice and the littermate controls; accompanied by 3D reconstructions of trabeculae from (A5) male control, (A7) male IFT88cKO, (A6) female control, and (A8) female IFT88cKO mice. **Dynamic Histomorphometry Analysis of Distal Femur Trabeculae in OsxCre; IFT88<sup>LoxP/LoxP</sup> mice was done for**

(B1) Trabecular BV/TV, (B2) Trabecular MAR, (B3) Trabecular MS/BS, and (B4) Trabecular BFR/BS. Measurements of trabecular MAR and BFR of the female IFT88cKO mice was significantly lower than that of the control females. Photo examples show trabecular fluorescent labeling in (B5) male control, (B7) male IFT88cKO, (B6) female control, and (B8) female IFT88cKO mice. TRAP-stained thin trabecular sections of the distal femur of IFT88cKO mice were compared to those of the controls. No significant difference was observed in any of the measured parameters, including (C1) Osteoclast number proportional to bone surface and (C2) Osteoclast surface proportional to bone surface. Photo examples show TRAP stain in (C3) male control, (C5) male IFT88cKO, (C4) female control, and (C6) female IFT88cKO mice. Note that *: p<0.05, ***: p<0.001, ****: p<0.0001.

Mechanical testing of the femur in three-point bending test showed no differences in structural mechanical properties between femurs of IFT88cKO and the littermate control groups (Fig 4). Young's modulus was significantly higher in the male IFT88cKO group compared to the control group. No other significant differences were observed.

Cortical analysis of femoral midshafts found no significant difference between the littermate controls and IFT88cKO mice in bone volume, bone surface, or bone area. It was observed, however, that both the male and female IFT88cKO groups showed a non-significant trend towards a decreased bone area when compared to their controls (S1 Fig.).

## Primary cilia in osteocytes play an important role in mechanically induced bone formation in MKS5cKO mice

A significant decrease in body weight between male control and male MKS5cKO mice was observed (male control 28.75g±1.13641, male MKS5cKO 26.3625g±1.197542, p=0.0011). There was no difference observed within the female groups, with their average body weights and standard deviations as follows: female control 24.64286±2.053336, female MKS5cKO 23.38333±1.725592 (Fig 5A). Furthermore, male MKS5cKO mice exhibited a trend toward shorter femurs than control mice but did not reach a level of statistical significance (p=0.0537) (Figs 5B & C1). No difference was observed within the female groups (Figs 6B & C2). BMD and BMC were analyzed using PIXImus. No significance was observed in BMD and BMC of either group (Figs 6D & E).

Trabecular analysis of μCT scans showed no significant differences in BV/TV in either male or female groups. Female MKS5cKO mice had a significant decrease in Tb.N (p=0.0207) and Tb.S (p=0.0497) when compared to female control mice, as well as a significant increase in Tb.Sp (p=0.0313). Male MKS5cKO mice exhibited significantly decreased Tb.Th (p=0.0419) compared to their control controls (Fig 6).

Histological analysis of the thin trabecular sections revealed a significant difference in BV/TV between the littermate control and MKS5cKO females (p=0.008). Although it did not reach significance, male conditional knockout mice had a BV/TV that was 17.93% lower than that of the control males. Even without reaching significance, a trend was noticed in all other parameters (MAR, MS/BS, and BFR/BS) in which the cKO values were lower than that of the control mice for both sexes (Fig 6). These data suggest that primary cilia in osteocytes influence bone formation in trabecular bone.

TRAP staining positive osteoclasts were counted on thin sections of the distal femur of all the animals. No significances in Oc.N/BS and Oc.S/BS were observed when compared between the littermate control and MKS5cKO groups (**Figure 7C1-6**). However, the number of osteoclasts present on stained thin sections of the male MKS5cKO mouse was 7.73% higher than that of the male control mouse, and for the female MKS5cKO, it was 1.05% higher than the female wild type. For Oc.S/BS, male MKS5cKO mice had an increase of 11.85% compared to the control males, while female MKS5cKO mice had an increase of 4% compared to the control females. These data support recent findings that primary cilia appear to regulate osteoclastogenesis [26].

Analysis of the three-point bending test showed no significant differences in most parameters measured (Fig 7). Total work required was significantly less for male MKS5cKO group compared to their sex-matched control, and although the average of the female MKS5cKO group showed a lower trend compared to the female control group, no significance was reached. Both male and female MKS5cKO groups also showed a trend towards lower toughness when compared to their respective controls, although no significance was reached in that parameter either.

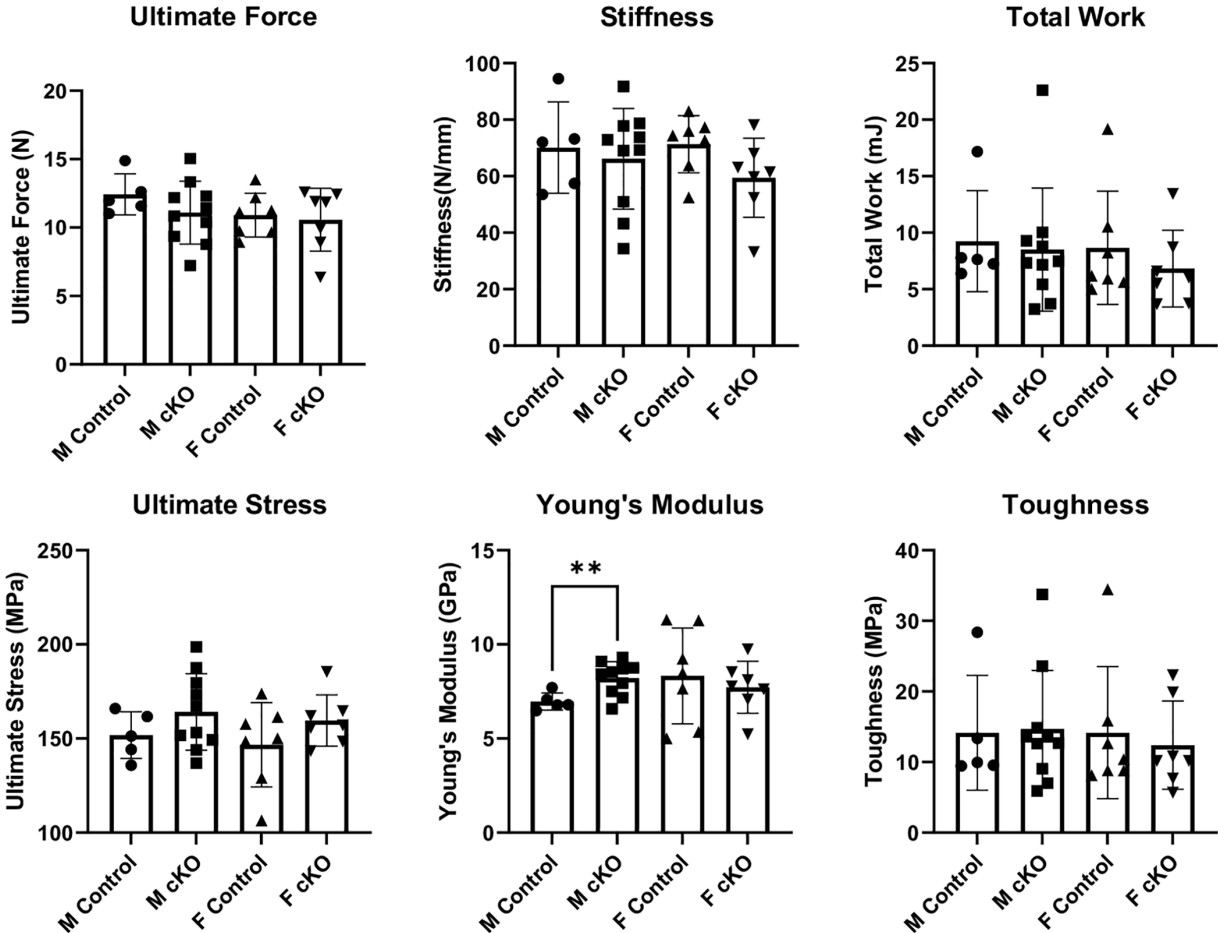

**Fig 4. Three-Point bending analysis of femoral mechanical properties in Osx-Cre; IFT88$^{LoxP/LoxP}$ mice.** Mechanical properties of femora from male and female Osx-Cre, IFT88$^{LoxP/LoxP}$ (IFT88cKO) mice and their littermate controls were assessed by three-point bending. Structural-level parameters included ultimate force, stiffness, and total work to failure. Material-level properties were evaluated by calculating ultimate stress, Young's modulus, and toughness. Male and female control and IFT88cKO groups are shown separately to assess sex-specific effects of IFT88 deletion. A significant increase in Young's modulus was observed in male IFT88cKO mice compared to male controls, indicating increased material stiffness of cortical bone. No significant differences were detected in ultimate force, stiffness, total work, ultimate stress, or toughness between control and IFT88cKO mice in either sex. Individual data points represent single animals, bars represent mean±SD. Note that **: $p < 0.01$.

Cortical analysis of μCT scans found no significant difference between the littermate controls and MKS5cKO mice in any of the measured parameters, including bone volume, bone surface, bone marrow area, and bone area (**S2 Fig**.).

To examine load-induced bone formation in MKS5 cKO Mice, histological analysis of the cortical sections of both left (non-loaded) and right (loaded) ulna was performed. When comparing bone formation parameters between right loaded and left non-loaded ulna, mechanical loading significantly increased load-induced bone formation in loaded ulna of both male and female MKS5 cKO mice and the littermate controls (**Table 1**). To determine load induced bone formation between MKS5cKO mice and the littermate controls, relative MS/BS (rMS/BS), relative MAR (rMAR) and relative BFR/BS (rBFS/BS) were calculated. Significant differences in rMS/BS and rBFR/BS between male and female control and MKS5cKO were observed (**Fig 8**). The rMAR parameter only revealed a significant difference between male the control and MKS5cKO mice ($p = 0.0058$), although a slight, though non-significant, difference was observed between the female control and MKS5cKO groups. Within the male groups, the conditional knockout mice had a MS/BS ($p = 0.0064$) that was

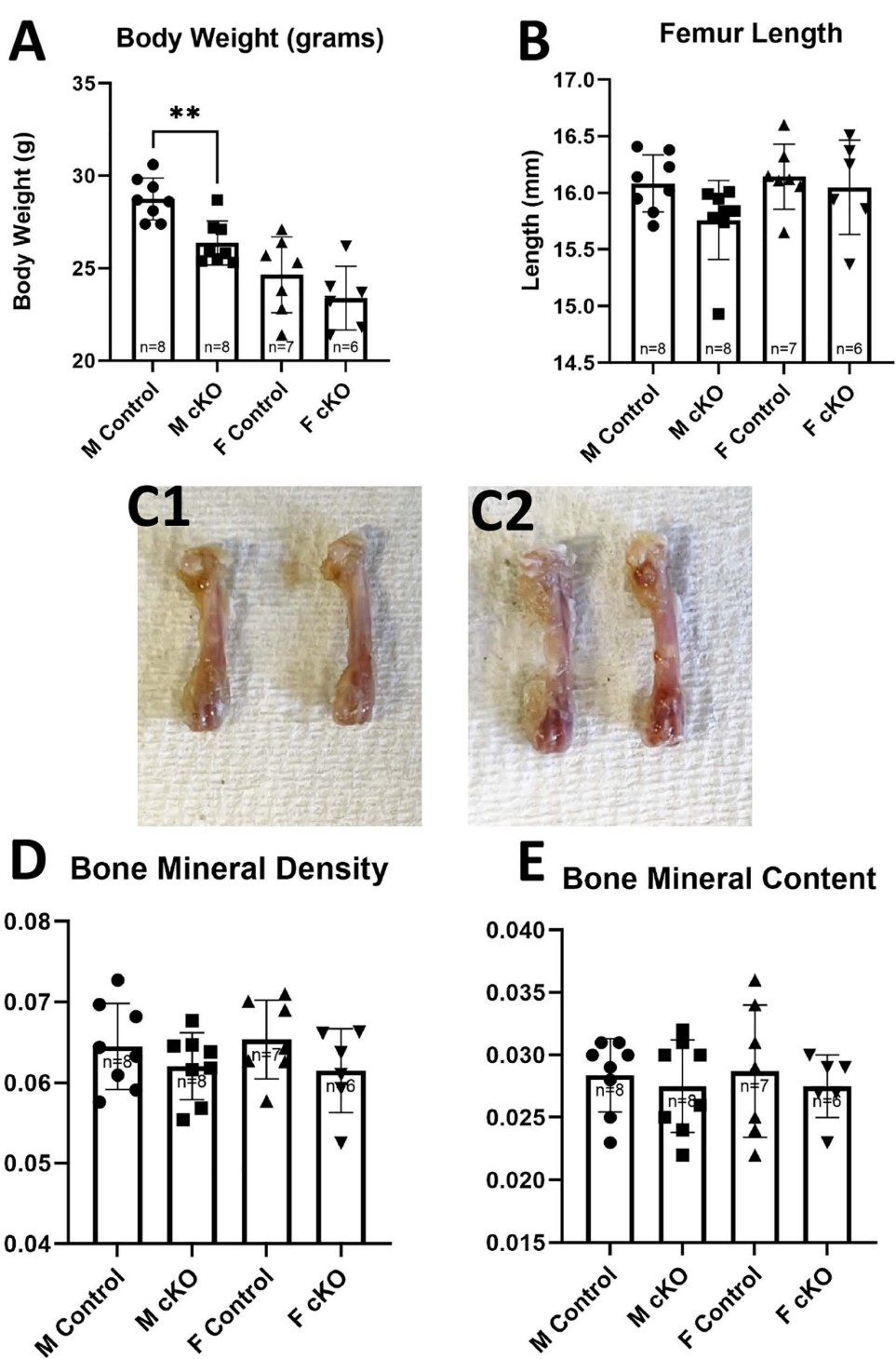

**Fig 5. Body weight, Length of femur, BMD and BMC of Dmp1-Cre; MKS5$^{LoxP/LoxP}$ mice. (A)** Body weights and (B) length of femur of the control and MKS5cKO mice at 20 weeks of age. Male MKS5cKO mice had significantly lower body weights compared to the male controls. (C1) Photo of femur of male MKS5cKO (Left) and control (Right) mice. (C2) Photo of femur of female MKS5cKO (Left) and control (Right) mice. Values for **(D)** BMD and **(E)** BMC. PIXImus scans of femurs reveal no difference in either BMD or BMC of MKS5cKO animals compared to the control mice. Note that **: $p < 0.01$.

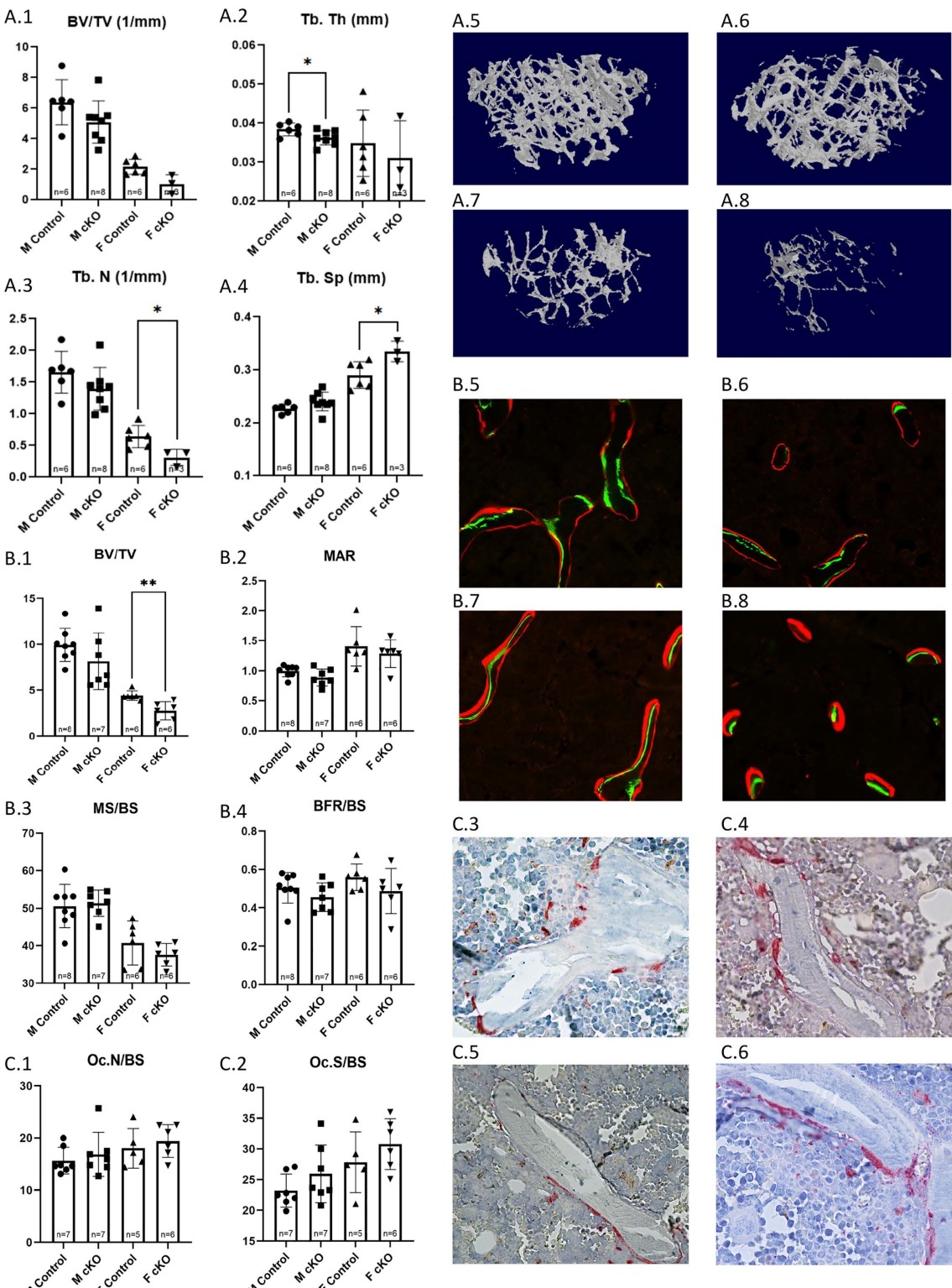

**Fig 6. Trabecular analysis of Dmp1-Cre+, MKS5LoxP/LoxP Mice.** (A1-4) Trabecular analysis results for distal femur of male and female MKS5cKO mice and the littermate controls; accompanied by 3D reconstructions of trabeculae from (A5) male WT, (A7) male MKS5cKO, (A6) female WT, and (A8) female MKS5cKO mice. **Dynamic Histomorphometry Analysis of Distal Femur Trabeculae in Dmp1-Cre; MKS5LoxP/LoxP** was done for (B1) Trabecular

BV/TV, (B2) Trabecular MAR, (B3) Trabecular MS/BS, and (B4) Trabecular BFR/BS. Measurements of trabecular BV/TV of the female MKS5cKO mice was significantly lower than that of the female controls (p = 0.008). Photo examples show trabecular fluorescent labeling in (B5) male control, (B7) male MKS5cKO mice, (B6) female control, and (B8) female MKS5cKO mice. TRAP-stained thin trabecular sections of the distal femur of MKS5cKO mice were compared to those of the controls. No significant difference was observed in any of the measured parameters, including (C1) Osteoclast number proportional to bone surface and (C2) Osteoclast surface proportional to bone surface. Photo examples show TRAP stain in (C3) male control, (C5) male MKS5cKO, (C6) female control, and (C8) female MKS5cKO mice. Note that *: p < 0.05, **: p < 0.01.

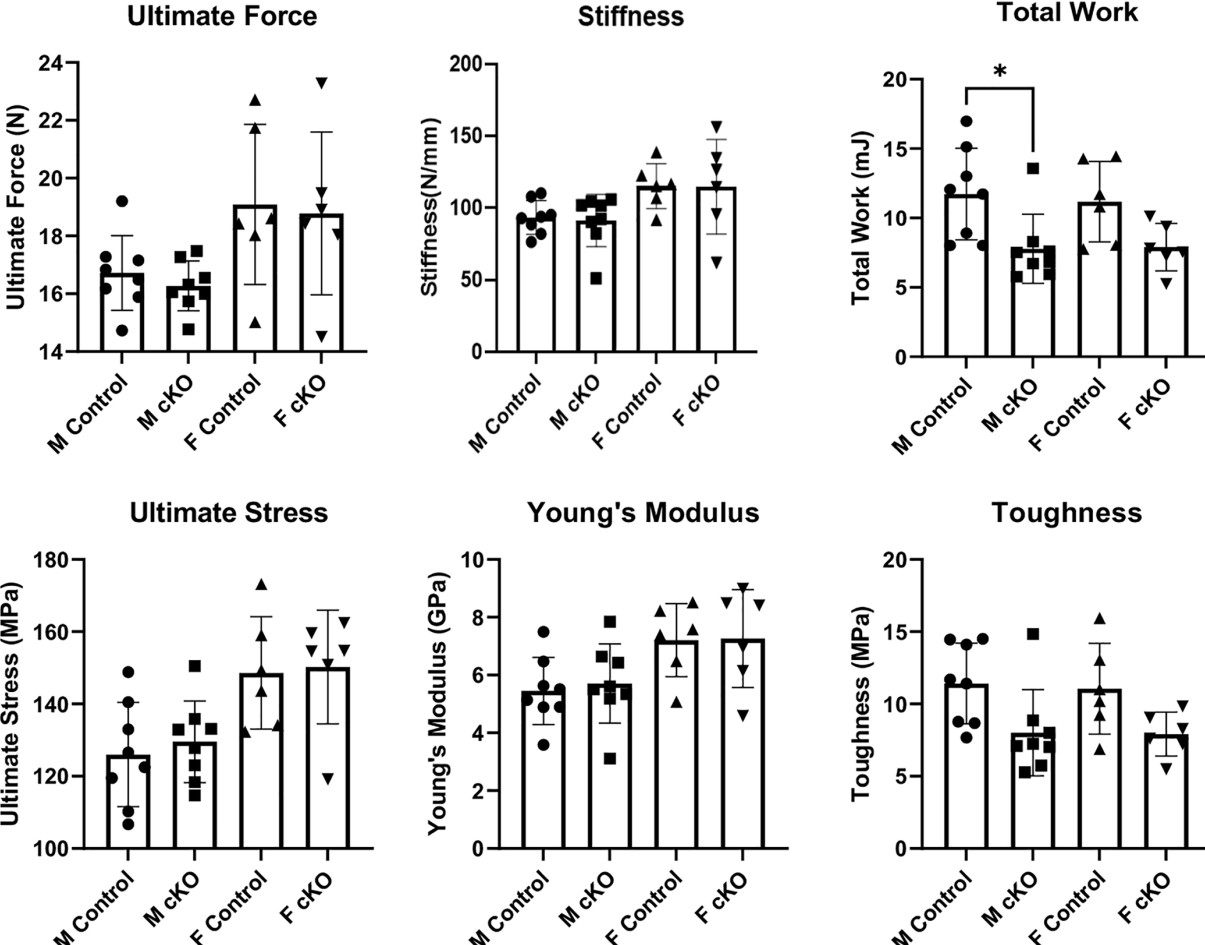

**Fig 7. Three-Point bending analysis of femoral mechanical properties in Dmp1-Cre; MKS5LoxP/LoxP mice.** Femoral mechanical properties of male and female Dmp1-Cre; MKS5LoxP/LoxP (MKS5cKO) mice and their littermate controls were assessed by three-point bending at 20 weeks of age. Structural mechanical parameters included ultimate force, stiffness, and total work to failure. Material-level properties were determined by calculating ultimate stress, Young's modulus, and toughness. Male and female control and MKS5cKO groups are shown separately to evaluate sex-specific effects of MKS5 deletion in osteocytes. A significant reduction in total work to failure was observed in male MKS5cKO mice compared to male controls, indicating reduced energy absorption capacity prior to fracture. No significant differences were detected in ultimate force, stiffness, ultimate stress, Young's modulus, or toughness between control and MKS5cKO mice in either sex. Individual data points represent single animals, and bars represent mean ± SD. Note that *: p < 0.05.

24.88% lower than that of the control group, as well as a MAR (0.0058) that was 46.27% lower and BFR/BS (p = 0.0026) that was 48.24% lower. For the female groups, the MS/BS (p = 0.0022) was 52.5% lower for the conditional knockout group compared to the control. The MAR was 27.58% lower, and the BFR/BS (p = 0.0325) was 41.54% lower when

**Table 1. Loading Induced Bone Formation Parameters at Ulnae.**

| Groups | n | MAR (µm/d) | | MS/BS (%) | | BFR/BS (µm3/µm2·yr) | |
|---|---|---|---|---|---|---|---|
| | | Mean±SEM | P value | Mean±SEM | P value | Mean±SEM | P value |
| **Male WT** | | | | | | | |
| Right | 8 | 1.162±0.07 | <0.0001 | 68.95±2.21 | <0.0001 | 0.81±0.09 | <0.0001 |
| Left | | 0.402±0.04 | | 21.31±2.74 | | 0.07±0.01 | |
| **Male MKS5cKO** | | | | | | | |
| Right | 7 | 0.654±0.04 | 0.0002 | 57.41±4.97 | <0.0001 | 0.368±0.025 | <0.0001 |
| Left | | 0.358±0.04 | | 20.27±3.36 | | 0.071±0.012 | |
| **Female WT** | | | | | | | |
| Right | 7 | 0.969±0.09 | 0.0002 | 57.29±4.14 | <0.0001 | 0.566±0.08 | 0.0005 |
| Left | | 0.312±0.08 | | 17.74±2.15 | | 0.0628±0.02 | |
| **Female MKS5cKO** | | | | | | | |
| Right | 6 | 0.877±0.09 | 0.0144 | 49.01±2.91 | 0.0007 | 0.431±0.05 | 0.003 |
| Left | | 0.400±0.13 | | 30.22±2.59 | | 0.137±0.05 | |

compared to the littermate control group (Fig 8). These data suggest primary cilia in osteocytes play an important role in load induced bone formation.

## Discussion

This study examined the mechanosensory role of primary cilia in bone growth and adaptation by examining two cilia specific genes, IFT88 and MKS5, required for proper cilia assembly and function.

This study demonstrates that the osteoblast primary cilium plays a role in bone development and growth. Both male and female IFT88 cKO mice were visibly smaller than their littermate controls, exhibiting significantly lower body weights and shorter femur lengths at eight weeks of age. Despite these pronounced phenotypes, DXA scans revealed no significant differences in BMC and BMD between cKO and the control mice. The high variability within the cKO groups may have contributed to the lack of significance in BMD measurements.

In trabecular bone, total bone volume remained unchanged in cKO mice compared to the littermate controls. However, female cKO mice showed a significant reduction in MAR and BFR when compared to their littermate controls. While males displayed a similar trend, the difference did not reach statistical significance, likely due to the greater variability among male mice. MS/BS values did not differ significantly between cKO and the control animals. While osteoclast numbers did not differ between either male or female cKO and the control groups, these data suggest elimination of primary cilia in osteoblasts only affects osteoblast activity. Further studies are required to assess which molecular pathways are affected by lack of primary cilia in osteoblasts.

µCT analysis of femoral cortical bone showed no significant differences; however, both male and female cKO groups exhibited lower averages in cortical bone volume, surface, and area compared to the littermate controls. In addition, ultimate force and total work were not significantly different between either male or female cKO and the controls. These data suggest that primary cilia did not significantly affect development of cortical bone during skeletal development.

The skeletal phenotypes observed in IFT88 cKO mice align with the findings from similar studies. In one study, deletion of the primary cilia using Prx1-Cre and IFT88$^{LoxP/LoxP}$ mice, which targets early limb bud mesenchyme, resulted in decreased long bone length. The proposed mechanisms included defects in endochondral bone formation, altered osteoblast differentiation and/or proliferation [27,28]. Another study, which deleted primary cilia by targeting IFT88 with Col2a-Cre, a marker of chondrocyte differentiation, also reported a significant reduction in body size between cKO compared

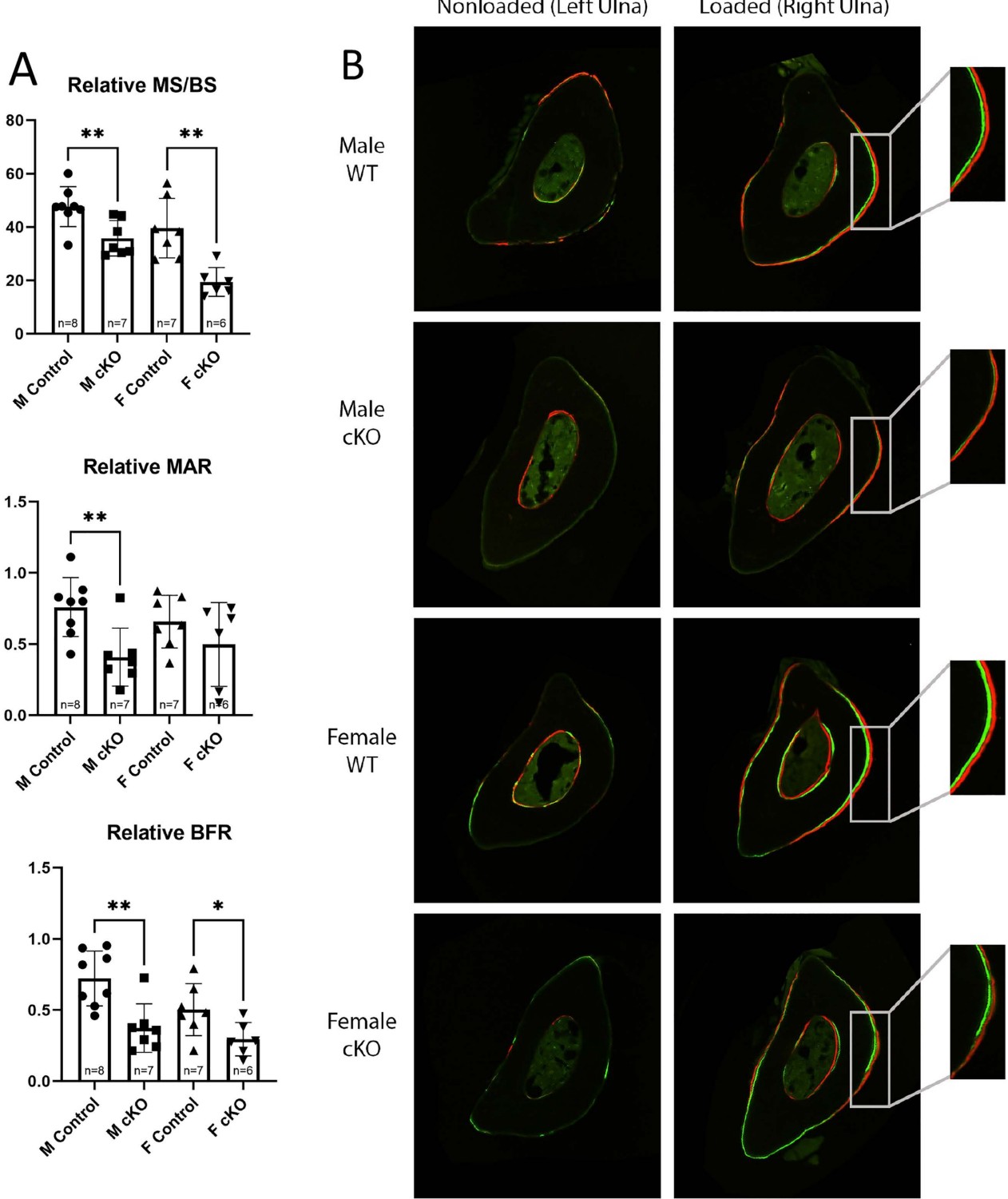

**Fig 8. Loading-Induced Bone Formation of Dmp1-Cre, MKS5$^{LoxP/LoxP}$ (cKO) Mice. (A)** The relative values, calculated as the difference between the right (loaded) and left (non-loaded) ulna was calculated in order to determine bone formation activity in the ulna as a result of mechanical loading. **(B)** Photo examples of cortical sections of right and left ulna from all four test groups. Note that *: $p < 0.05$, **: $p < 0.01$.

to controls [27,29]. Additionally, research using the same Cre recombinase to delete IFT20, rather than IFT88, identified notable skeletal phenotypes, including a significant decrease in BMD of their cKO mice [28].

Collectively, these finds support the conclusion that primary cilia in osteoblast are essential for bone growth. Given their role in coordinating multiple signaling pathways within the skeletal system, primary cilia represent a promising target for therapeutic strategies aimed at treating bone loss diseases.

The results of this study indicate that the osteocyte primary cilium plays a crucial role in bone homeostasis and mechanotransduction, supporting the hypothesis that primary cilia function as mechanosensors in bone cells. MKS5 cKO mice appeared smaller than their littermate control counterparts, with a significant reduction in body size observed in males. While females exhibited a similar trend, the difference did not reach statistical significance. At 20 weeks of age, male cKO mice has a significantly lower body weights than the controls and showed a tendency toward shorter femur lengths. Despite lower body weights and a seemingly smaller femur, no significant differences were detected in BMD and BMC in either sex.

In trabecular sections, the female MKS5 cKO group exhibited a significant decrease in BV/TV compared to the female controls. Although MS/BS, MAR, and BFR were lower in female cKO mice, these differences did not reach significance. Similarly, male cKO mice displayed lower BV/TV, MAR, and BFR/BS than male control mice, but none of these differences were statistically significant. No significant differences were observed in osteoclast number or osteoclast surface. These data suggest that dysfunctional primary cilia have a greater impact one bone homeostasis in female mice than in males.

Dynamic ulnar loading showed a striking impairment in mechanotransduction in MKS5 cKO in this study. Both male and female MKS5 cKO mice exhibited significantly lower relative MS/BS and relative BFR than the controls, indicating a diminished response to mechanical loading. The decreased rBFR and rMAR following loading in MKS5 cKO mice, resulting from primary cilia align with previous studies demonstrating that osteocyte primary cilia are essential for load-induced bone formation [16]. Several studies have investigated the role of primary cilia in osteocyte mechanotransduction, including one that examined osteoblast- and osteocyte-specific deletion of Kif3a, a key component of the IFT anterograde motor Kinesin-2. In that study, Kif3a cKO mice exhibited significantly reduced bone formation after mechanical loading compared to controls [30,31]. However, because their model affected both osteoblasts and osteocytes, it was unclear whether the observed effects were due to cilia-deficient osteoblasts or osteocytes. Additionally, Kif3a has known non-ciliary functions, making it difficult to attribute the results solely to its role within primary cilium. By using Dmp1-Cre as our recombinase, we achieved osteocyte-specific recombination. Deleting MKS5, a cilia-specific protein, in osteocytes resulting dysfunctional cilia and significantly impaired bone formation in response to loading. These findings further emphasize the critical role of primary cilia in osteocyte mechanotransduction.

This study highlights the essential role of primary cilia in bone health. Deletion of MKS5 and IFT88 in specific bone cells led to impaired mechanically induced bone formation and disrupted bone growth, respectively, demonstrating that primary cilia are crucial for bone mechnaotransduction and development. Mechanistically, ciliary dysfunction is likely to perturb multiple signaling pathways known to localize to or be regulated by the primary cilium. Wnt/β-catenin signaling, which is tightly coupled to osteoblast differentiation and bone anabolism, has been shown to be modulated by ciliary integrity and intraflagellar transport proteins [1,7,19]. Hedgehog signaling, a canonical cilium-dependent pathway essential for osteoblast lineage commitment and skeletal patterning, is also disrupted by loss of ciliary components such as IFT88 [1,13,19,27]. In osteocytes, ciliary dysfunction may further impair TGF-β signaling, which has been reported to localize to the ciliary base and regulate bone remodeling coupling [14,32]. In addition, primary cilia have been implicated in coordinating mechanosensitive ion channel activity, including Piezo1 and TRPV4, both of which mediate fluid flow-induced calcium signaling in osteocytes and are required for normal load-induced bone formation [33–35]. Together, these observations provide a mechanistic framework linking ciliary dysfunction to impaired skeletal mechano-adaptation and development.

A limitation of this study is the use of Osterix-Cre, which is not strictly osteoblast-specific and is also active in subsets of growth plate chondrocytes [36]. This is particularly relevant to the reduced femur length observed in IFT88 mutant mice, as longitudinal bone growth is primarily regulated by chondrocyte proliferation and hypertrophy during endochondral ossification. Disruption of primary cilia in chondrocytes is known to impair cilium-dependent pathways such as Hedgehog signaling, which are essential for normal growth plate function [29,37,20]. Therefore, the shortened femur length may partially reflect ciliary dysfunction in chondrocytic lineage cells rather than osteoblast-lineage effects alone.

Future studies should evaluate mice at various time points, from birth through adulthood and aging, to determine whether the observed differences in bone development and adaptation persist, diminish, or are attenuated over time. Diaphanization of newborn mice could provide further insights into the effects of primary cilia deletion on skeletal development. Another important area for further exploration is the influence of sex hormones on primary cilia formation and functions, particularly their role in mechanical loading (e.g., exercise). Additionally, investigating how cilia knockout affects the proliferation, differentiation, and communication of specific bone cell types would offer valuable insights.

In conclusion, this study underscores the critical role of primary cilia in bone development and mechano-adaptation. The findings suggest the functional primary cilia in osteoblasts are essential for skeletal development, while those in osteocytes play a key role in mechanically induced bone formation, highlighting its potential as a therapeutic target in bone loss prevention.

## Supporting information

**S1 Fig. Cortical bone μCT of IFT88 cKO mice.**
(PDF)

**S2 Fig. Cortical bone μCT of MKS5 cKO mice.**
(PDF)

## Acknowledgments

We would like to thank the Indiana Center for Musculoskeletal Health (ICMH) Histology Core for assisting with histological preparation. We also thank Staci Engle and Ruchi Bansal for their technical support.

## Author contributions

**Conceptualization:** Mariana Moraes de Lima Perini, Nicolas F Berbari, Jiliang Li.

**Data curation:** Mariana Moraes de Lima Perini, Alyssa F Fayemi, Julie N Pugh, Elizabeth M Scott, Karan Bhula, Austin Chirgwin, Olivia N White.

**Formal analysis:** Mariana Moraes de Lima Perini, Alyssa F Fayemi, Julie N Pugh, Elizabeth M Scott, Karan Bhula, Austin Chirgwin, Olivia N White.

**Funding acquisition:** Jiliang Li.

**Investigation:** Mariana Moraes de Lima Perini, Alyssa F Fayemi, Julie N Pugh, Elizabeth M Scott, Karan Bhula, Austin Chirgwin, Olivia N White.

**Methodology:** Mariana Moraes de Lima Perini, Alyssa F Fayemi, Julie N Pugh, Elizabeth M Scott, Olivia N White, Nicolas F Berbari, Jiliang Li.

**Project administration:** Jiliang Li.

**Resources:** Jiliang Li.

**Validation:** Alyssa F Fayemi, Nicolas F Berbari.

**Visualization:** Mariana Moraes de Lima Perini, Alyssa F Fayemi.

**Writing – original draft:** Mariana Moraes de Lima Perini, Alyssa F Fayemi, Jiliang Li.

**Writing – review & editing:** Mariana Moraes de Lima Perini, Alyssa F Fayemi, Julie N Pugh, Elizabeth M Scott, Karan Bhula, Austin Chirgwin, Olivia N White, Nicolas F Berbari, Jiliang Li.

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
