## [Decision Letter · Decision Letter 0]

23 Nov 2025

Dear Dr. Li,

Thank you for submitting your manuscript to PLOS ONE. After careful consideration, we feel that it has merit but does not fully meet PLOS ONE’s publication criteria as it currently stands. Therefore, we invite you to submit a revised version of the manuscript that addresses the points raised during the review process.

We look forward to receiving your revised manuscript.

Kind regards,

Toshio Matsumoto

Academic Editor

PLOS ONE

Journal Requirements:

2. To comply with PLOS One submissions requirements, in your Methods section, please provide additional information regarding the experiments involving animals and ensure you have included details on (1) methods of sacrifice, (2) methods of anesthesia and/or analgesia, and (3) efforts to alleviate suffering.

3. Please include your tables as part of your main manuscript and remove the individual files. Please note that supplementary tables (should remain/ be uploaded) as separate "supporting information" files.

“We would like to thank the Indiana Center for Musculoskeletal Health (ICMH) Histology Core for assisting with histological preparation. We also thank Staci Engle and Ruchi Bansal for their technical support. This project was supported by NIH R21 AR074012 (JL).”

7. When completing the data availability statement of the submission form, you indicated that you will make your data available on acceptance. We strongly recommend all authors decide on a data sharing plan before acceptance, as the process can be lengthy and hold up publication timelines. Please note that, though access restrictions are acceptable now, your entire data will need to be made freely accessible if your manuscript is accepted for publication. This policy applies to all data except where public deposition would breach compliance with the protocol approved by your research ethics board. If you are unable to adhere to our open data policy, please kindly revise your statement to explain your reasoning and we will seek the editor's input on an exemption. Please be assured that, once you have provided your new statement, the assessment of your exemption will not hold up the peer review process.

8. PLOS ONE now requires that authors provide the original uncropped and unadjusted images underlying all blot or gel results reported in a submission’s figures or Supporting Information files. This policy and the journal’s other requirements for blot/gel reporting and figure preparation are described in detail at https://journals.plos.org/plosone/s/figures#loc-blot-and-gel-reporting-requirements and https://journals.plos.org/plosone/s/figures#loc-preparing-figures-from-image-files. When you submit your revised manuscript, please ensure that your figures adhere fully to these guidelines and provide the original underlying images for all blot or gel data reported in your submission. See the following link for instructions on providing the original image data: https://journals.plos.org/plosone/s/figures#loc-original-images-for-blots-and-gels.

Additional Editor Comments:

As indicated by the two expert reviewers, the authors should explain the rationale and background for deleting different genes, IFT88 for ciliogenesis and MKSS for ciliary gating function, for different cell differentiation stages, osteoblasts and osteocytes, respectively. It is also of critical importance to clarify what the actual control mice are by stating “WT mice”. If the authors actually used WT mice instead of Cre or floxed mice, it should not be a good control for studying mice with conditional gene deletion.

There are other important concerns and criticisms raised by the two reviewers, which should be properly addressed by the authors.

Reviewers' comments:

Reviewer's Responses to Questions

**Comments to the Author**

1. Is the manuscript technically sound, and do the data support the conclusions?

Reviewer #1: Yes

Reviewer #2: Partly

2. Has the statistical analysis been performed appropriately and rigorously?

Reviewer #1: Yes

Reviewer #2: Yes

3. Have the authors made all data underlying the findings in their manuscript fully available?

Reviewer #1: Yes

Reviewer #2: No

4. Is the manuscript presented in an intelligible fashion and written in standard English?

Reviewer #1: Yes

Reviewer #2: Yes

Reviewer #1: General Comments

This manuscript presents a well-designed and thoughtfully executed study examining the roles of primary cilia in osteoblasts and osteocytes using Osx-Cre;IFT88^fl/fl^ and Dmp1-Cre;MKS5^fl/fl^ mouse models. The experiments are technically sound, the results are internally consistent, and the overall findings provide valuable insights into how primary cilia contribute to skeletal development and mechanically induced bone formation. The study meaningfully advances our understanding of cilia-mediated mechanobiology in bone.

At the same time, I believe the manuscript could be further strengthened by clarifying several important conceptual and methodological points. One issue concerns the genotype of the control mice. The manuscript refers to the control group simply as “WT,” but it is unclear whether these animals were true wild-type mice or Cre-negative; flox/flox littermates. Because Cre drivers such as Osx-Cre and Dmp1-Cre can independently influence skeletal phenotype, and floxed alleles may not be strictly equivalent to wild-type alleles, it is generally important that conditional knockout studies use appropriate littermate controls matched for both Cre status and floxed alleles. If such controls were used—which is likely—a brief clarification in the Methods section would greatly improve transparency.

Additionally, because the study uses two different ciliary genes (IFT88 and MKS5) in two different cell types, the rationale behind these choices could be explained more explicitly. A concise description of how IFT88 functions in ciliogenesis and how MKS5 regulates ciliary gating at the transition zone would help readers understand how these complementary models interrogate distinct aspects of ciliary biology across the osteoblast–osteocyte lineage. Addressing these points would enhance accessibility for readers who are less familiar with ciliary molecular mechanisms.

Major Points

1. The study applies IFT88 deletion (abolishing ciliogenesis) to osteoblasts and MKS5 deletion (impairing ciliary function) to osteocytes, but the rationale behind using different levels of ciliary disruption in different cell types is not explicitly stated. Providing a brief explanation—biological, technical, or both—would greatly improve conceptual clarity. Readers would benefit from knowing why complete loss of cilia was examined in osteoblasts, while functional impairment was examined in osteocytes.

2. To deepen the conceptual understanding of the study, I would like to ask the authors to consider the following two points:

What phenotype would occur if MKS5 were deleted in osteoblasts?

(Cilia would remain structurally intact but functionally impaired, potentially resulting in a milder defect in bone formation compared with IFT88 deletion.)

What phenotype would occur if IFT88 were deleted in osteocytes?

(Complete loss of ciliogenesis would likely lead to a more pronounced reduction in load-induced bone formation compared with MKS5 deletion.)

Adding this conceptual discussion will help clarify whether the observed phenotypes reflect gene-specific properties, cell-type specificity, or differentiation-stage dependence, thereby making the overall logic of the study more robust.

3. The manuscript would benefit from a concise discussion of possible signaling pathways affected by ciliary dysfunction—such as Wnt/β-catenin, Hedgehog, TGF-β, or mechanosensitive channels (Piezo1, TRPV4)—to strengthen the mechanistic interpretation.

4. As noted above, please specify whether the control animals were Cre-negative; flox/flox littermates rather than true wild-type mice. This clarification is important given the known effects of Cre drivers on bone biology.

Minor Points

1. Clearly specify the statistical tests used and report exact p-values when possible.

2. Ensure consistent labeling and significance notation across the figures.

Addressing the points above—particularly the rationale for gene selection, clarification of control genotypes, and the requested conceptual examination of reversed gene–cell combinations—will significantly improve the clarity and impact of the manuscript.

Reviewer #2: MAJOR POINTS

There are two main criticisms to address here:

1) This reviewer cannot understand why the authors knocked out two different molecules (IFT88 and MKS5) to disrupt functional cilia. Are the authors sure that deletion of either gene results in the same impact on cilia (i.e., total loss of cilia and/or its function)? If so, the authors should provide experimental evidence or published facts supporting the idea. If not, it is hard to compare the two different animals.

2) The affected cell lineages in the cKO animals are unclear. For example, in the IFT88cKO mice using the Osx-Cre, chondrocytes as well as osteocytes can also be affected. So, osteoblasts may not be responsible for all the observed phenotypes including growth defect. The authors should analyze all the cell lineage for possible changes in cilia in the animals used.

3) The authors should reorganize the whole logic after the two points above are clarified. Otherwise, it would be impossible to interpret the results properly.

MINOR POINTS

1) There are no page numbers.

2) Table 1 was not found.

3) The picture in Figure 1 B is not well focused and of poor quality, which should be replaced with a better one.

.

Reviewer #1: **Yes:** Masahiro HiasaMasahiro HiasaMasahiro HiasaMasahiro Hiasa

Reviewer #2: No

---

## [Author Response · Author response to Decision Letter 1]

13 Jan 2026

We would like to thank the editors and reviewers for their time and effort that have put into improving our manuscript. We have addressed all the comments as detailed on our point-by-point response below. For clarity, we first list the reviewers’ comments, our responses are underneath each comment highlighted in yellow background.

Reviewer #1:

One issue concerns the genotype of the control mice. The manuscript refers to the control group simply as “WT,” but it is unclear whether these animals were true wild-type mice or Cre-negative; flox/flox littermates. Because Cre drivers such as Osx-Cre and Dmp1-Cre can independently influence skeletal phenotype, and floxed alleles may not be strictly equivalent to wild-type alleles, it is generally important that conditional knockout studies use appropriate littermate controls matched for both Cre status and floxed alleles. If such controls were used—which is likely—a brief clarification in the Methods section would greatly improve transparency.

We used Cre-positive and flox/flox negative mice as the littermate controls. We deleted “WT” but used littermate control or the controls in the text and figures.

Additionally, because the study uses two different ciliary genes (IFT88 and MKS5) in two different cell types, the rationale behind these choices could be explained more explicitly.

We add a short paragraph in the Introduction to justify two different ciliary genes knockout mice in this study on Page 6.

A concise description of how IFT88 functions in ciliogenesis and how MKS5 regulates ciliary gating at the transition zone would help readers understand how these complementary models interrogate distinct aspects of ciliary biology across the osteoblast–osteocyte lineage. Addressing these points would enhance accessibility for readers who are less familiar with ciliary molecular mechanisms.

a brief description of IFT88 and MKS5 functions was added in the Introduction on Page 6.

Major Points

1. The study applies IFT88 deletion (abolishing ciliogenesis) to osteoblasts and MKS5 deletion (impairing ciliary function) to osteocytes, but the rationale behind using different levels of ciliary disruption in different cell types is not explicitly stated. Providing a brief explanation—biological, technical, or both—would greatly improve conceptual clarity. Readers would benefit from knowing why complete loss of cilia was examined in osteoblasts, while functional impairment was examined in osteocytes.

IFT88 deletion in osteocytes has been done by a graduate student in Dr. Jacobs Lab in 2015. We have IFT88 and MKS5 floxed mice available. So, we did ITF88 in osteoblasts and MKS5 in osteocytes. We add a description in Introduction on Page 6.

2. To deepen the conceptual understanding of the study, I would like to ask the authors to consider the following two points:

What phenotype would occur if MKS5 were deleted in osteoblasts?

(Cilia would remain structurally intact but functionally impaired, potentially resulting in a milder defect in bone formation compared with IFT88 deletion.)

We agree with you if mice with MKS5 deletion in osteoblasts would result in a milder defect in bone formation compared with IFT88. However, for unknown reason, breeding between osterix-cre and MKS5 floxed mice did not go well. Because we want to focus on mechanotransduction, we focused on osteocytes.

What phenotype would occur if IFT88 were deleted in osteocytes?

(Complete loss of ciliogenesis would likely lead to a more pronounced reduction in load-induced bone formation compared with MKS5 deletion.) Adding this conceptual discussion will help clarify whether the observed phenotypes reflect gene-specific properties, cell-type specificity, or differentiation-stage dependence, thereby making the overall logic of the study more robust.

IFT88 deletion in osteocytes significantly impairs load-indued bone formation by 47% based on the graduate thesis by Dr. An M. Nguyen from Christopher Jacobs published in 2015. However, it is difficult to compare which mouse strain would lead to a more pronounced reduction in loading induced bone formation if the load-induced bone formation is not completely suppressed.

3. The manuscript would benefit from a concise discussion of possible signaling pathways affected by ciliary dysfunction—such as Wnt/β-catenin, Hedgehog, TGF-β, or mechanosensitive channels (Piezo1, TRPV4)—to strengthen the mechanistic interpretation.

Thank you for these suggestions. A concise discussion was added on Page 21. Actually, we have started working on some of those signaling molecules using live cell imaging.

4. As noted above, please specify whether the control animals were Cre-negative; flox/flox littermates rather than true wild-type mice. This clarification is important given the known effects of Cre drivers on bone biology.

Please see our response above to your general comment.

Minor Points

1. Clearly specify the statistical tests used and report exact p-values when possible.

All the exact p values were reported in the text.

2. Ensure consistent labeling and significance notation across the figures.

Thanks for the reminder. We double checked the labeling and significance notation.

Reviewer #2:

There are two main criticisms to address here:

1) This reviewer cannot understand why the authors knocked out two different molecules (IFT88 and MKS5) to disrupt functional cilia. Are the authors sure that deletion of either gene results in the same impact on cilia (i.e., total loss of cilia and/or its function)? If so, the authors should provide experimental evidence or published facts supporting the idea. If not, it is hard to compare the two different animals.

IFT88 deletion suppresses ciliogenesis. Then there is not primary cilia on the cells. IFT88 deletion in osteocytes has been done by a graduate student in Dr. Jacobs Lab in 2015. So, we did ITF88 in osteoblasts. Without MSK5, abnormal primary cilia can form on the cells. We add a description in Introduction to justify the use of two genes knockout mouse model to examine the function of primary cilia in skeletal development and mechanotransduction on page 6.

2) The affected cell lineages in the cKO animals are unclear. For example, in the IFT88cKO mice using the Osx-Cre, chondrocytes as well as osteocytes can also be affected. So, osteoblasts may not be responsible for all the observed phenotypes including growth defect. The authors should analyze all the cell lineage for possible changes in cilia in the animals used.

The reviewer is right. Osx-cre may affect hypertrophic chondrocytes. But in this study, we used osx-cre as the littermate controls. Both cKO mice and the control mice contain osx-cre gene. Therefore, the phenotypic differences between cKO mice the control mice were caused by either IFT88 cKO or MKS5 cKO.

3) The authors should reorganize the whole logic after the two points above are clarified. Otherwise, it would be impossible to interpret the results properly.

Additional justifications were added in the Introduction on Page 6. In conclusion, functional primary cilia are necessary for skeletal development and load-induced bone formation.

MINOR POINTS

1) There are no page numbers.

Page numbers were added.

2) Table 1 was not found.

Corrected.

3) The picture in Figure 1 B is not well focused and of poor quality, which should be replaced with a better one.

A set of new images were used to replace the old ones.

---

## [Decision Letter · Decision Letter 1]

27 Jan 2026

Dear Dr. Li,

Thank you for submitting your manuscript to PLOS ONE. After careful consideration, we feel that it has merit but does not fully meet PLOS ONE’s publication criteria as it currently stands. Therefore, we invite you to submit a revised version of the manuscript that addresses the points raised during the review process.

We look forward to receiving your revised manuscript.

Kind regards,

Toshio Matsumoto

Academic Editor

PLOS One

**Journal Requirements:**

Reviewers' comments:

Reviewer's Responses to Questions

**Comments to the Author**

Reviewer #1: (No Response)

Reviewer #2: (No Response)

2. Is the manuscript technically sound, and do the data support the conclusions?

Reviewer #1: Yes

Reviewer #2: Yes

3. Has the statistical analysis been performed appropriately and rigorously?

Reviewer #1: Yes

Reviewer #2: N/A

4. Have the authors made all data underlying the findings in their manuscript fully available?

Reviewer #1: Yes

Reviewer #2: No

5. Is the manuscript presented in an intelligible fashion and written in standard English?

Reviewer #1: Yes

Reviewer #2: Yes

Reviewer #1: The authors have responded to the previous comments in an appropriate and careful manner, and the overall clarity and conceptual coherence of the manuscript have been substantially improved. However, with respect to the mechanical testing data shown in Figures 4 and 7 (three-point bending), there is still room for improvement in terms of figure self-containment.

To ensure reproducibility and to facilitate accurate interpretation by readers, the Figure legends for Figures 4 and 7 should explicitly state the sample size (n) for each group, the statistical methods used, the definition of statistical significance or the exact p values, as well as the units of each mechanical parameter and a brief clarification of what each parameter represents. In addition, a typographical error in the term stiffness is present in Figure 7 and should be corrected.

These issues represent relatively minor revisions and do not affect the conclusions of the study; however, addressing them would substantially improve the completeness and interpretability of the figures. Furthermore, for the other figures in the manuscript, the statistical analysis methods and the use (or absence) of post hoc tests should be clearly stated either in the Methods section or in the corresponding Figure legends.

Reviewer #2: Revised manuscript has significantly been improved. However, one critical point remains unsolved. In the comment 2) in the previous review, I asked the authors to analyze all the cell lineage for possible changes in cilia (or at least IFT88 expression) in the animals used. The authors seem to have misunderstood the question. Because osterix expression is not "osteoblast-specific," the authors should examine primary cilia in bone cells, at least in cells of the chondrocyte lineage known to express osterix. Osteocytes can also be affected. If IFT88 gene has also been deleted in chondrocytic lineage cells, osteoblasts may not be totally responsible for the observed phenotype. Accordingly, the frequently used phrase "osteoblast-specific" should be softened in the text, and the leaky nature of osterix promoter should be explicitly described as a limitation.

.

Reviewer #1: No

Reviewer #2: No

---

## [Author Response · Author response to Decision Letter 2]

4 Mar 2026

We would like to thank the editors and reviewers again for their time and effort that have put into improving our manuscript. We have addressed all the comments as detailed on our point-by-point response below. For clarity, we first list the reviewers’ comments, our responses are underneath each comment highlighted in yellow background.

Reviewer #1:

To ensure reproducibility and to facilitate accurate interpretation by readers, the Figure legends for Figures 4 and 7 should explicitly state the sample size (n) for each group, the statistical methods used, the definition of statistical significance or the exact p values, as well as the units of each mechanical parameter and a brief clarification of what each parameter represents. In addition, a typographical error in the term stiffness is present in Figure 7 and should be corrected.

These issues represent relatively minor revisions and do not affect the conclusions of the study; however, addressing them would substantially improve the completeness and interpretability of the figures. Furthermore, for the other figures in the manuscript, the statistical analysis methods and the use (or absence) of post hoc tests should be clearly stated either in the Methods section or in the corresponding Figure legends.

We would like to thank the reviewer for these suggestions. We fixed the error in Figure 7, and add additional information in Statics and legends of Figures 4 & 7.

Reviewer #2: Revised manuscript has significantly been improved. However, one critical point remains unsolved. In the comment 2) in the previous review, I asked the authors to analyze all the cell lineage for possible changes in cilia (or at least IFT88 expression) in the animals used. The authors seem to have misunderstood the question. Because osterix expression is not "osteoblast-specific," the authors should examine primary cilia in bone cells, at least in cells of the chondrocyte lineage known to express osterix. Osteocytes can also be affected. If IFT88 gene has also been deleted in chondrocytic lineage cells, osteoblasts may not be totally responsible for the observed phenotype. Accordingly, the frequently used phrase "osteoblast-specific" should be softened in the text, and the leaky nature of osterix promoter should be explicitly described as a limitation.

We agree that Osterix expression is not strictly osteoblast-specific and that Cre activity has been reported in chondrocyte lineage cells and, to a lesser extent, osteocytes. We apologize for misunderstanding the reviewer’s original question and appreciate the opportunity to clarify this point.

Importantly, our interpretation is supported by prior work demonstrating that IFT88 deletion in Prx1- or Osterix-expressing cells impairs bone formation and mechanosensitive responses, consistent with a critical role for primary cilia in osteoblast-lineage cells. Nevertheless, we agree that the observed phenotype may reflect combined effects of ciliary dysfunction in multiple Osterix-expressing lineages rather than osteoblasts alone.

In response to the reviewer’s comment, we have revised the manuscript to discuss the potential contribution of chondrocytes and osteocytes to the phenotype in the Discussion section at Page 21. We believe these revisions provide a more accurate and appropriately cautious interpretation of our findings.

---

## [Decision Letter · Decision Letter 2]

16 Mar 2026

Primary cilia in osteoblasts and osteocytes are required for skeletal development and mechano-adaptation

PONE-D-25-58640R2

Dear Dr. Li,

We’re pleased to inform you that your manuscript has been judged scientifically suitable for publication and will be formally accepted for publication once it meets all outstanding technical requirements.

Kind regards,

Toshio Matsumoto

Academic Editor

PLOS One

**Comments to the Author**

Reviewer #1: All comments have been addressed

Reviewer #2: All comments have been addressed

2. Is the manuscript technically sound, and do the data support the conclusions?

Reviewer #1: Yes

Reviewer #2: (No Response)

3. Has the statistical analysis been performed appropriately and rigorously?

Reviewer #1: Yes

Reviewer #2: (No Response)

4. Have the authors made all data underlying the findings in their manuscript fully available?

Reviewer #1: Yes

Reviewer #2: (No Response)

5. Is the manuscript presented in an intelligible fashion and written in standard English?

Reviewer #1: Yes

Reviewer #2: (No Response)

Reviewer #1: The revisions have improved the presentation and interpretability of the data. I have no further major concerns and believe the manuscript is suitable for publication in its current form.

Reviewer #2: (No Response)

.

Reviewer #1: No

Reviewer #2: No

---

## [Editor Report · Acceptance letter]

PONE-D-25-58640R2

PLOS One

Dear Dr. Li,

I'm pleased to inform you that your manuscript has been deemed suitable for publication in PLOS One. Congratulations! Your manuscript is now being handed over to our production team.

Kind regards,

on behalf of

Dr. Toshio Matsumoto

Academic Editor

PLOS One